# How Cancer Exploits Ribosomal RNA Biogenesis: A Journey beyond the Boundaries of rRNA Transcription

**DOI:** 10.3390/cells8091098

**Published:** 2019-09-17

**Authors:** Marco Gaviraghi, Claudia Vivori, Giovanni Tonon

**Affiliations:** 1Experimental Imaging Center; Ospedale San Raffaele, 20132 Milan, Italy; 2Functional Genomics of Cancer Unit, Division of Experimental Oncology, Istituto di Ricovero e Cura a Carattere Scientifico (IRCCS) San Raffaele Scientific Institute, 20132 Milan, Italy; 3Centre for Genomic Regulation (CRG), The Barcelona Institute for Science and Technology, 08003 Barcelona, Spain; claudia.vivori@crg.eu; 4Center for Translational Genomics and Bioinformatics, Istituto di Ricovero e Cura a Carattere Scientifico (IRCCS) San Raffaele Scientific Institute, 20132 Milan, Italy

**Keywords:** ribosomal RNA (rRNA), ribosomes, oncogenes, cancer, small nucleolar RNAs (snoRNA), decapping

## Abstract

The generation of new ribosomes is a coordinated process essential to sustain cell growth. As such, it is tightly regulated according to cell needs. As cancer cells require intense protein translation to ensure their enhanced growth rate, they exploit various mechanisms to boost ribosome biogenesis. In this review, we will summarize how oncogenes and tumor suppressors modulate the biosynthesis of the RNA component of ribosomes, starting from the description of well-characterized pathways that converge on ribosomal RNA transcription while including novel insights that reveal unexpected regulatory networks hacked by cancer cells to unleash ribosome production.

## 1. Introduction

Since the dawn of the 20th century, pathologists have noticed that nucleoli were prominently enlarged and increased in number in cancer cells [1,2]. Further studies demonstrated how this phenotype was linked at the molecular level to hyperactivated transcription of ribosomal DNA (rDNA, reviewed in [3,4]). Nucleoli, the largest subnuclear organelles, are membrane-less and highly dynamic structures mainly devoted to the synthesis and processing of ribosomal RNA (rRNA) and to the assembly of ribosomal subunits. Nucleoli are structured around specific DNA sequences called nucleolar organizer regions (NORs), distributed along the short arms of all human acrocentric chromosomes [5,6]. Starting from the NORs, the human nucleolus assembles in three different subnucleolar compartments, characterized by different protein compositions and accommodating the diverse steps of ribosome biosynthesis. The first and most internal region of the nucleolus, which directly surrounds rDNA sequences, is called the fibrillar center (FC), where transcription of most rRNA species is performed. Juxtaposed to the FC is the dense fibrillar component (DFC), where subsequent cleavage and post-transcriptional modification of rRNA precursors (pre-rRNA) occur. The last and outer nucleolar shell is represented by the granular component (GC), where rRNA maturation is completed and mature rRNA assembles with ribosomal proteins, giving rise to ribosomal subunits that will subsequently be exported in the cytoplasm [7,8]. However, nucleoli are far from being uniform structures and their size and structural features are highly dynamic and vary as a function of the rRNA biosynthesis rate [9]. rRNA and ribosome synthesis is an extremely energy-demanding process: the production of a functional ribosome requires the engagement of the entire set of cellular RNA polymerases to produce rRNA and messenger RNAs (mRNA) coding for ribosomal proteins and a large number of diverse accessory proteins comprising endo and exoribonucleases, ATP-dependent RNA helicases, chaperones, or assembly factors, and a vast array of ribonucleoprotein complexes [10,11]. Thus, although ribosome biogenesis is generally perceived as a steady, house-keeping process, it is instead finely regulated, promptly responding to the various cellular conditions and energy supplies [12]. As such, the mis-regulation of the ribosomal biogenesis pathway is associated with the development of a large number of diseases, including cancer [3,13].

In this review, we will explore how cancer cells hack rRNA biogenesis to boost the activity of the cellular translation machinery. We will summarize the established roles of oncogenes and tumor suppressor genes in the regulation of rRNA transcription and will also introduce new pieces of evidence pointing to a novel role of rRNA processing in driving oncogenesis.

## 2. rRNA Transcription and Cancer

One of the main biological functions hosted inside nucleoli consists in the biogenesis of three of the four mature rRNA molecules. rDNA cluster genes are transcribed by RNA polymerase I (PolI), giving rise to a polycistronic 47S pre-rRNA, containing the 18S, 5.8S, and 28S rRNA sequences separated by internal and external transcribed spacers (called ITS1, ITS2 and 5′-ETS, 3′-ETS, respectively). Like all polymerases, PolI is a holoenzyme whose core proteins interact with a series of auxiliary factors. These proteins are involved in the recognition of rDNA promoter sequences, in the binding to rDNA, and in the initiation, elongation, and termination of 47S pre-rRNA transcription [14]. rRNA transcription initiates with the recruitment and the formation of a pre-initiation complex (PIC) on rDNA promoter sites. One of the main factors that participate in PIC formation is the protein upstream binding factor (UBF), which wraps around rDNA promoter sequences by interacting with the DNA minor groove [15,16]. Another important component of PIC acting in synergy with UBF is the selectivity factor SL1, a protein complex containing the TATA-binding protein (TBP) and three PolI-specific TBP-associated factors; TAF_I_48, TAF_I_68, and TAF_I_95/110 [17,18]. The physical tethering of the PolI catalytic core to the PIC is then mediated by the recruitment of PolI-associated factor 53 (PAF53) on the UBF platform. Subsequently, the transcription initiation factor TIF-IA, a PolI accessory subunit, binds to SL1, allowing to precisely position the polymerase on the rDNA promoter sequence [14,19].

The release of the TIF-IA factor elicits the transition of PolI status from initiation to transcriptional elongation. TIF-IA, when not engaged in a PIC, is kept unphosphorylated and unable to bind to PolI. During the PIC formation, TIF-IA is phosphorylated to promote PolI tethering, but is promptly dephosphorylated as soon as PolI starts transcribing, allowing for its recycling [20]. Eventually, terminator elements on both sides of the repetitive transcription units drive proper transcription termination. Indeed, these sequences are bound by the transcription terminator factor TTF-I, a specific DNA binding protein that in turn stops PolI-mediated transcriptional elongation [21].

Targeting the PolI molecular machinery represents the most established mechanism exploited by cancer cells to fuel the ribosomal biosynthesis process. While gain of function mutations affecting PolI components are not frequent and are mostly associated with hereditary diseases rather than with cancer [22,23], tumor cells boost rRNA biogenesis mainly through the overexpression of genes involved in PolI-mediated transcription and through the stimulation of their activity, which is directly controlled by oncogenic signaling pathways. Indeed, several of the most frequently altered pathways in cancer impact on almost any of the proteins required for PIC formation or for transcriptional elongation and termination (Figure 1). Most importantly, the transcription factors UBF, SL1, and TIF-IA undergo extensive post-translational modifications (PTMs) driven by the oncogenic kinases of the RAS mitogen-activated protein kinases (RAS-MAPK) and mTOR pathways [24,25,26,27].

For example, UBF shows enhanced phosphorylation in fast growing cells compared to quiescent ones [28]. Indeed, both cyclin/cell cycle-dependent kinase (CDK) complexes and MAPKs phosphorylate UBF, hence stimulating rRNA transcription by increasing UBF-PolI binding [29,30]. Moreover, mTOR drives the phosphorylation of the carboxy-terminal portion of UBF boosting PolI-mediated rRNA transcription [31]. Similarly, acetylation on UBF has also been related to PolI hyper-activation. UBF is acetylated by nucleolar-residing histone trans-acetylases in a cell cycle-dependent fashion leading to increased rDNA transcription [32,33]. On the other hand, UBF phosphorylation by casein kinase 2 (CK2) has a detrimental effect on rRNA transcription as it impairs the ability of UBF to recruit and activate PolI (summarized in Figure 1A) [34]. PTMs have also been described for the SL1 complex. Indeed, SL1 subunit TAF_I_68 presents different states of acetylation correlated with PolI transcription rates [35].

Another crucial regulation step of PolI activity is represented by PTMs of TIF-IA, which are induced by extracellular growth signals through the mTOR-S6 kinase and RAS-MAPKs axes [14] and can have different outcomes on rRNA transcription. Inhibition of mTOR signaling inactivates TIF-IA by decreasing phosphorylation at S44 and enhancing phosphorylation at S199 [26,36]. Moreover, two additional serine residues (S633 and S649) are targeted by ERK and RSK protein kinases respectively, thus increasing PolI transcription and ribosome biogenesis in response to serum stimulation [37]. On the other hand, TIF-IA is targeted by inhibitory phosphorylations that halt rRNA biogenesis in stress conditions. For example, c-Jun n-terminal protein kinase 2 (JNK2), activated in many conditions of cellular stress, inhibits PolI transcriptional activity through the phosphorylation of TIF-IA T200 residue [25]. Another inhibitory phosphorylation event that ultimately shuts down rDNA transcription in condition of energy shortage is triggered by a high intracellular ratio of AMP/ATP. This unbalance activates the AMP-activated protein kinase (AMPK) that directly phosphorylates TIF-IA on S635, hampering its binding to the SL1 complex and thus its ability to bridge the PIC-PolI interaction (Figure 1A) [38].

Besides inducing stimulatory PTMs on rRNA transcription-related proteins, oncogene activation can also affect the global expression levels of proteins belonging to PolI machinery. The activation of the RAS pathway in colon cancer upregulates TBP, which in turn promotes the tethering of PolI to rDNA promoters and increases rRNA transcription (Figure 1B) [39,40].

A prominent role in the regulation of rRNA transcription in cancer is played by the C-MYC oncogene (summarized in Figure 1B). C-MYC boosts all steps of rRNA biosynthesis and maturation through diverse molecular mechanisms. The first evidence of an involvement of C-MYC in rRNA biogenesis came from the observation that C-MYC localizes inside nucleoli when overexpressed in cell lines [41]. Subsequently, it was demonstrated that C-MYC increases 47S pre-rRNA synthesis through its direct recruitment on rDNA. Indeed, C-MYC binds ribosomal gene loci through its consensus sequences (E-boxes) located on the rDNA gene promoter. Once recruited on rDNA, C-MYC stimulates PolI transcription by interacting with the TBP and TAFs subunits of SL1, as well as with the TRRAP histone acetyltransferase co-factor that is responsible for the increase of histone acetylation levels on rDNA chromatin [42,43]. C-MYC also increases the levels of the independently transcribed 5S rRNA molecule by promoting RNA polymerase III (PolIII) transcription through its binding to the TFIIIB transcription initiation factor [44]. Finally, C-MYC indirectly upregulates rRNA biosynthesis by stimulating the RNA polymerase II (PolII)-mediated transcription of several ribosomal proteins and of a “PolI regulon”, which consists of mRNAs encoding for factors involved in PolI-mediated transcription [45,46]. Of note, genes encoding for UBF and TIF-IA transcription factors, as well as for the PolI enzyme itself, all feature an E-box and are C-MYC transcriptional targets [46,47].

Beyond the stimulatory effect of oncogenes on rRNA biogenesis, several tumor suppressor proteins negatively target PolI transcription in order to control ribosome generation (Figure 1B, right panel). The frequent loss of these tumor suppressors observed in cancer thus represents an additional mechanism by which tumor cells boost rRNA synthesis and cell growth. Among these, p53 prevents both PolI and PolIII activation by interfering with PolI PIC formation and UBF-SL1 binding [48] and by disrupting the interaction of two of the core PolIII transcription factors by binding to TFIIIB and thus decreasing PolIII recruitment [49]. Furthermore, p53 is intimately linked to nucleolar integrity as this tumor suppressor is the main hub of the nucleolar stress response. Indeed, a wide variety of cellular stresses induce nucleolar disruption and a consequent release of ribosomal proteins outside the nucleolus. Once in the nucleoplasm, RPL5, RPL11, and RPL23 ribosomal proteins, as well as 5S rRNA, interact and sequester HDM2, ultimately stabilizing p53 and promoting its activation (reviewed in [12,50]).

In addition to p53, other tumor suppressor proteins that are frequently lost in cancer modulate rRNA and nucleolar dynamics. For example, the retinoblastoma (Rb) transcriptional repressor accumulates in the nucleolus of confluent cells and decreases PolI activity by binding to UBF and impairing its recruitment on rDNA [51,52]. A prominent localization inside the nucleolar granular component was also detected for p14^ARF^ [53] and, although controversies have emerged regarding its role in the nucleolus related to p53 activation [54], few reports point at a role of the ARF tumor suppressor in inhibiting rDNA transcription. p14^ARF^ was shown to interact with rDNA promoter sequences and its exogenous expression anti-correlated with UBF phosphorylation and led to the reduction of 47S pre-rRNA transcription [55,56]. Moreover, p14^ARF^ blocks rDNA transcription by interacting with the TTF-I termination factor, displacing it from the nucleolus by interfering with its binding with the nucleolar granular component protein nucleophosmin (NPM) [57].

On the contrary, PTEN phosphatase prevents PolI activation by interfering with SL1 incorporation in the PIC, both in a PI3K-Akt-mTOR-dependent and independent fashion [58]. Moreover, recent evidences show that a N-terminal extended isoform of PTEN named PTENβ is predominantly localized inside nucleoli, where it inhibits rRNA transcription [59].

## 3. rRNA Processing and Cancer

After being transcribed, the 47S pre-rRNA undergoes a maturation process whereby mature rRNA molecules are generated. The polycistronic 47S transcript is progressively subjected to a series of endo- and exonucleolytic cleavages taking place inside ITS and ETS sequences (Figure 2). As a result, the 47S pre-rRNA is initially divided into a shorter molecule precursor of 18S rRNA, which will be included in the small ribosome subunit (SSU), and a longer pre-rRNA containing both the 28S and 5.8S rRNA, forming the large ribosome subunit (LSU), which are subsequently processed to release fully matured rRNAs (reviewed in [60,61]). Maturing rRNAs are also extensively post-transcriptionally modified by ribonucleoprotein complexes, composed by a class of noncoding RNA molecules called small nucleolar RNAs (snoRNAs), which interact with several structural and enzymatic proteins, giving rise to small nucleolar ribonucleoproteins (snoRNPs, reviewed in [62,63]).

Whereas the pleiotropic effects of oncogenes and tumor suppressors on rRNA transcription have been well documented, less is known about the impact of cancer-related genes on the regulation of rRNA maturation and processing (summarized in Figure 2).

A genome-wide study initially shed light on the possible contribution of C-MYC oncogene in promoting rRNA processing. Gene expression analysis performed in a human C-MYC Tet-OFF B-cell line and in rat primary fibroblasts expressing the MYC-estrogen receptor (MYC-ER) fusion protein (whose nuclear localization is triggered by tamoxifen administration) identified several genes belonging to the rRNA processing pathway as common C-MYC-regulated genes in both experimental systems [64]. Among the C-MYC targets genes identified in this study, two of them, *WDR12* [65] and *NIFK* [66], were shown to promote the biogenesis of the LSU by participating in the maturation of both 28S and 5.8S rRNAs (Figure 2). WDR12 is a component of the nucleolar trimeric complex PeBoW, in combination with PES1 and BOP1, which participates in the processing of 32S pre-rRNA [67]. NIFK, instead, was initially reported as a nucleolar protein interacting with the Ki-67 proliferation marker [68]. As proof of their role in LSU biogenesis, the silencing of *WDR12* or *NIFK* or the expression of mutant forms unable to interact respectively with their protein partners or with ITS2 sequences in pre-rRNA molecules had dramatic effects on 28S and 5.8S rRNA maturation, resulting in the accumulation of 32S pre-rRNA. Moreover, silencing or mutating *WDR12* or *NIFK* induced nucleolar stress that led to p53 stabilization, consequently halting cell proliferation [65,66].

A contribution of C-MYC in promoting SSU biogenesis was instead suggested by its interplay with the UTP14a protein, which was shown to act on 18S rRNA maturation [69]. Indeed, a recent work demonstrated that UTP14a is a C-MYC transcriptional target and interestingly described an additional feed-forward loop, by which UTP14a interacts directly with C-MYC, stabilizes it, and prevents its degradation [70]. Moreover, UTP14a was also shown to promote p53 destabilization, further sustaining cancer progression (Figure 2) [69].

Similarly, in the context of N-MYC-driven neuroblastomas, MYC oncogene was shown to control the transcription of the *DEF*/*UTP25* gene, whose encoded protein participates in the proper processing of 18S rRNA species [71,72]. As a result, UTP25 synergizes with N-MYC to foster tumor development both in Zebrafish and in human cells, while its haploinsufficiency reduces the growth of N-MYC-driven neuroblastoma cells [73]. Curiously, *UTP25* haploinsufficiency did not cause alterations of 28S/18S rRNA ratios, expected as a consequence of the 18S rRNA processing block, while only the complete ablation of *UTP25* resulted in the accumulation of 18S rRNA precursors. In line with these observations, it was demonstrated that the overexpression of both N-MYC and UTP25 mediates the switch to an alternative rRNA processing route named “pathway 2” to generate mature rRNA species (Figure 2). Interestingly, rRNA processing pathway 2 is preferentially utilized by aggressive cancers as a faster route to generate mature rRNA molecules [60]. In all, these evidences suggest that rewiring the rRNA maturation pathway represents an additional strategy exploited by oncogenes, such as MYC, to speed up ribosome biogenesis in cancer.

In addition to MYC, other oncoproteins such as several transducers of mitogenic signals, impact on rRNA maturation. Indeed, both AKT and mTOR pathways were shown to affect rRNA processing rates in addition to their established function in promoting PolI-dependent rRNA transcription (Figure 2) [74,75]. Instead, the role of the C-JUN transcription factor in rRNA processing is more characterized. Indeed, C-JUN partially localizes inside the nucleolus, where it stabilizes the binding of DDX21 RNA helicase to pre-rRNA molecules, fostering rRNA maturation (Figure 2) [76].

As for tumor suppressors, few studies performed in mouse fibroblasts suggest that nucleolar ARF protein could also participate in the regulation of rRNA processing. Indeed, the artificial expression of p19^ARF^ in NIH-3T3 cells delayed the maturation of 28S and 18S rRNAs, causing the accumulation of aberrantly processed rRNA precursors as well as a block in proliferation, which depends on the ability of p19^ARF^ to establish interactions with 5.8S rRNA and NPM [77,78]. Interestingly, the ectopic expression of MDM2 results in the disruption of the NPM-p19^ARF^ interaction, ultimately relieving the p19^ARF^-induced proliferation block [79]. These data thus suggest that the p19^ARF^-dependent inhibitory effect on rRNA processing might be hampered in tumor cells, by reducing ARF expression, as well as through the overexpression of MDM2 [80,81].

## 4. Alterations of snoRNA Dynamics in Cancer

snoRNAs are a family of more than 200 unique single-stranded non-coding RNAs which foster rRNA processing. These RNAs catalyze RNA editing reactions through the formation of specific base pairs with their target rRNAs. Based on conserved nucleotide motifs, snoRNAs are divided in C/D box and H/ACA box families: C/D box snoRNAs guide the 2′-*O*-ribose methylation of rRNAs, while H/ACA box snoRNAs direct the pseudo-uridylation of rRNAs and are generally larger compared to C/D box snoRNAs [82,83,84].

snoRNAs belonging to both families associate with core structural and catalytic proteins, forming snoRNPs. The enzymatic proteins that mediate rRNA editing reactions are specific for each snoRNA family. In particular, the enzyme that mediates C/D box snoRNA-dependent 2′-*O*-ribose methylation is fibrillarin (encoded by the *FBL* gene) while H/ACA snoRNAs rely on dyskerin (whose coding gene is *DKC1*) to promote rRNA pseudo-uridylation [85,86]. In humans, approximately 120 C/D box snoRNPs have been shown to catalyze the 2′-*O*-ribose methylation of 110 rRNA bases, while almost 90 H/ACA box snoRNPs drive 100 rRNA pseudo-uridylation reactions [87]. Besides their role in rRNA editing, specific snoRNAs are also directly involved in the processing events of pre-rRNA molecules. In particular, two major C/D box snoRNAs, *SNORD3* (U3) and *SNORD118* (U8), are required for the cleavage steps that lead to the release of mature 18S and 28S–5.8S rRNA, respectively [88,89]. In addition, *SNORD14* (U14), *SNORA73A* (U17/E1/snR30), and *SNORD22* (U22) were also shown to direct cleavages inside 5′-ETS and ITS1 spacer sequences [90,91,92].

Given the role of snoRNAs in the ribosome biogenesis pathway, alterations in their levels and function in cancer is expected. A comprehensive study interrogating The Cancer Genome Atlas database revealed that the aberrant expression of snoRNAs is a widespread feature of cancer cells, and a general upregulation of snoRNAs (in particular belonging to the C/D box class) was observed across different cancer types, associated with the ectopic expression of both fibrillarin and dyskerin catalytic enzymes [93]. This evidence suggests that cancer cells may require an accelerated biogenesis of both snoRNAs and snoRNA-associated proteins to sustain cell growth, likely by stimulating snoRNP-dependent rRNA editing. Indeed, although the precise roles of post-transcriptional modifications on rRNA molecules are still not completely understood, it was proposed that rRNA base modifications might increase rRNA half-life and regulate ribosome translation capacity (reviewed in [94,95]), mechanisms that could both positively affect cancer development. However, formal proofs that link rRNA editing to cancer development are still limited. In addition, it is increasingly emerging that snoRNAs can affect cancer cell growth not only by promoting modifications of rRNA bases, but also through novel extra-nucleolar functions. Indeed, some snoRNA molecules seem to have microRNA-like features since they can be processed by the RNA interference machinery to generate small RNA fragments (called sdRNAs) able to modulate gene expression as well as alternative splicing in cancer cells [96,97]. Therefore, some snoRNAs may behave both as oncogenes or tumor suppressors according to the cellular mechanism they regulate (reviewed in [98,99]).

Further corroborating snoRNA oncogenic activity, a specific signature of upregulated snoRNAs was identified in non-small cell lung cancer samples, including *SNORD33* (U33), *SNORD66* (HBII-142), *SNORD73B* (U73), *SNORD76* (U76), *SNORD78* (U78), and *SNORA42* (ACA42) [100]. On the other hand, one of the first examples of tumor suppressive snoRNAs was brought by the observation that the *SNORD50* (U50) gene residing in the 6q14.3 chromosomal locus is frequently lost in cancer. In prostate and breast cancer, loss of U50 expression is associated to the presence of either large deletions, affecting the 6q chromosomal region or more focal deletions inside the *SNORD50* gene [101,102]. In agreement with these evidences, the exogenous overexpression of U50 reduced the colony forming potential of both prostate and breast cancer cells, thus confirming its tumor suppressive activity [101,102]. Another evidence of tumor suppressive snoRNAs comes from a study that demonstrated how a class of snoRNAs used as a normalization control for gene expression analysis is instead frequently downregulated in cancer and associated with poor prognosis [103].

As for the transcription and processing of rRNA, the biogenesis of the snoRNP machinery is also directly stimulated by the action of oncogenes. Evidences from both *Drosophila* and human cancer cell lines show that a large number of snoRNAs are downregulated upon the depletion of the MYC oncogene and that several snoRNA host genes or genes encoding for snoRNPs feature MYC binding sites in their promoter sequences [104]. These data not only imply that the pleiotropic MYC stimulation of ribosome biogenesis pathway also includes a direct transcriptional control on snoRNA and snoRNA-related genes but might also suggest that snoRNAs can mediate and potentiate the downstream effects of oncogene activation leading to cancer establishment and progression. This is the case of acute myeloid leukemia (AML), where the interference with oncogene-induced snoRNA biogenesis was shown to attenuate the oncogenic potential of leukemic cells [105]. In this context, the AML1-ETO fusion oncoprotein stimulates the maturation of C/D box snoRNPs through the interaction between its downstream effector AES and DDX21. As a consequence, the depletion of snoRNAs in AML cells dramatically decreases rRNA 2′-*O*-ribose methylation, therefore reducing protein translation rates and ultimately impairing the oncogene-induced proliferation and self-renewal capacities of leukemic cells (Figure 3A).

Interestingly, this is not an AML1-ETO specific feature, as the overexpression of other oncogenes driving AML, including C-MYC, had the same outcome on snoRNPs biogenesis and leukemogenesis [105]. Moreover, snoRNAs and snoRNPs upregulation was associated with tumor development in several other contexts, such as in breast cancer [106] and in lung tumor-initiating cells [100]. These observations suggest that the snoRNA biosynthetic pathway represents another step exploited by oncogenes, with the final goal of fueling ribosome biogenesis. In this view, tumor cells do not only modulate snoRNAs involved in rRNA editing. On the contrary, it was demonstrated that both U3 and U8 snoRNAs, which have a crucial role in aiding the endonucleolytic cleavages required respectively for 18S and 28S-5.8S rRNAs maturation, can represent driving forces towards tumorigenesis. Indeed, U3 and U8 genetic depletion was shown to block the maturation of pre-rRNAs in a panel of cancer cell lines of different tissue origin, leading to a decrease in mature ribosome subunits [107]. As a consequence, the reduction of both U3 and U8 levels dramatically impaired cell proliferation and cancer development in xenograft models and was associated with defects in protein translation and in p53-dependent nucleolar stress [108].

Accordingly, it was shown that the U3 function can be promoted by putative oncoproteins, such as SIRT7, an enzyme belonging to the sirtuin family and found overexpressed in different cancer types, where it is associated with bad prognosis [109,110,111]. SIRT7 has pleiotropic roles in the regulation of rRNA biogenesis [112] and its expression can also affect snoRNA dynamics. Specifically, SIRT7 directly binds and stabilizes mature U3 snoRNA and promotes the de-acetylation of a component of the U3 snoRNP complex, U3-55k, enhancing its binding to U3 and fostering 18S rRNA maturation [113]. Interestingly, the interaction between U3-55k and U3 snoRNA was suggested to control U3 expression levels as U3-55k depletion, obtained both by RNA interference or subsequent to the differentiation of human colon and lung cell lines, resulted in the reduction of U3, possibly by preventing the accumulation of mature U3 snoRNPs (Figure 3B) [114].

U8 levels were also suggested to be regulated in cancer at the transcriptional level. Indeed, an inverse correlation between *SNORD118* expression and the methylation status of the genomic sequences upstream of *SNORD118* was detected throughout 24 different tumor types [93], likely indicating that the loss of *SNORD118* methylation drives its expression in cancer.

Besides the ability of oncogenes to regulate snoRNAs expression, more sophisticated molecular mechanisms controlling U3 and U8 snoRNAs processing and activity in cancer are recently emerging, involving specific features of the snoRNA molecule. U3 and U8, like other snoRNAs transcribed by PolII from independent genes, are protected by a 5′7-methylguanosine (m^7^G) cap that is hypermethylated to 2,2,7-trimethylguanosine (m_3_G) by the trimethylguanosine synthetase TGS1 during their maturation in their route to the nucleolus [115,116,117,118]. Intriguingly, studies in *Xenopus laevis* oocytes identified X29 as one of the proteins co-purifying with U8 snoRNA and described it as a RNA decapping enzyme capable of removing m_3_G cap structures from U8 snoRNA both in *in vitro* assays and in living cells [119,120]. As a result, decapped U8 showed a remarkable reduced half-life and led to the block of U8-dependent pre-rRNA processing [120], suggesting that snoRNA decapping represents a novel mechanism that negatively regulates snoRNA activity, ultimately impairing rRNA processing. A human orthologue of X29 exists, named NUDT16, which shares with X29 the ability to decap U8 molecules in vitro [121]. Interestingly, the *NUDT16* gene locus is hypermethylated and silenced in the context of T-cell lymphoblastic leukemia, and its loss is associated with the upregulation of C-MYC and with an increase of tumor growth rates [122]. However, while X29 is mainly localized inside nucleoli [119], NUDT16 has an almost complete cytoplasmic localization in human cells [123], raising questions regarding the conservation of the X29 snoRNA-decapping function from *Xenopus* to human.

In this regard, we have recently identified an entire novel nucleolar pathway regulating snoRNA decapping in human cells, prominently engaged in carcinogenesis [124]. Specifically, we found that the DCP1α/DCP2 decapping machinery, which surveys aberrant mRNAs in the cytoplasm (reviewed in [125]), is also tethered inside nucleoli by establishing direct interactions with the Proline-rich Nuclear Receptor Coactivator 1 (PNRC1) nucleolar tumor suppressive protein. Inside the nucleolus, the PNRC1-DCP1α/DCP2 complex interacts with U3 and U8 snoRNAs and promotes the removal of their m_3_G cap, thus delaying the processing of 47S pre-rRNA. We also demonstrated that this molecular mechanism is frequently hacked by cancer cells with the aim to stimulate ribosome biogenesis and cell growth (Figure 3B). Indeed, we and others have shown that *PNRC1* is generally expressed in normal cells but is pervasively deleted and down-regulated in cancer [124,126,127,128,129]. Moreover, the re-expression of wild-type PNRC1, but not of a mutant form unable to interact with the decapping complex and to promote U3 and U8 decapping, restrained the hyper-proliferation of cancer cell lines overexpressing either oncogenic RAS or C-MYC, thus suggesting that these oncogenes rely on functional U3 and U8 snoRNAs to sustain cancer progression.

Taken together, these studies shed light on a novel additional layer of regulation in the rRNA biogenesis pathway that exploits the activity of RNA decapping proteins on snoRNA molecules inside nucleoli. Curiously, this gatekeeping mechanism is likely conserved throughout evolution in eukaryotes, such as the *Saccharomyces cerevisiae* Edc2 RNA decapping co-activator, which shares with PNRC1 the ability to interact with DCP2-dependent decapping machinery, was described to partially localize inside nucleoli [130]. Therefore, it is reasonable to think that oncogenes may disrupt this checkpoint by interfering with the expression or the nucleolar localization of these decapping-related proteins. However, although pieces of evidence for promoter hypermethylation in cancer have emerged for both *NUDT16* and *PNRC1* [122,131], the molecular mechanisms by which oncogenes may regulate nucleolar decapping are still unknown.

## 5. Concluding Remarks

Cancer cells are extremely dependent on accelerated rates of rRNA and ribosome biogenesis, and the interference with these processes has been demonstrated to have strong detrimental effects on tumor development and growth. Thanks to the extensive understanding about the molecular mechanisms (mis)regulating rRNA transcription in cancer, compounds targeting the PolI transcriptional machinery are currently in clinical trials to treat different types of cancer [132,133,134,135]. However, novel interesting connections between cancer and different steps of rRNA processing are emerging, not only opening possibilities for the development of new anti-tumoral drugs, but also revealing new, largely unexplored pathways that add significant details to the extremely complicated and heterogeneous landscape of the ribosome biogenesis process.

## Figures and Tables

**Figure 1 cells-08-01098-f001:**
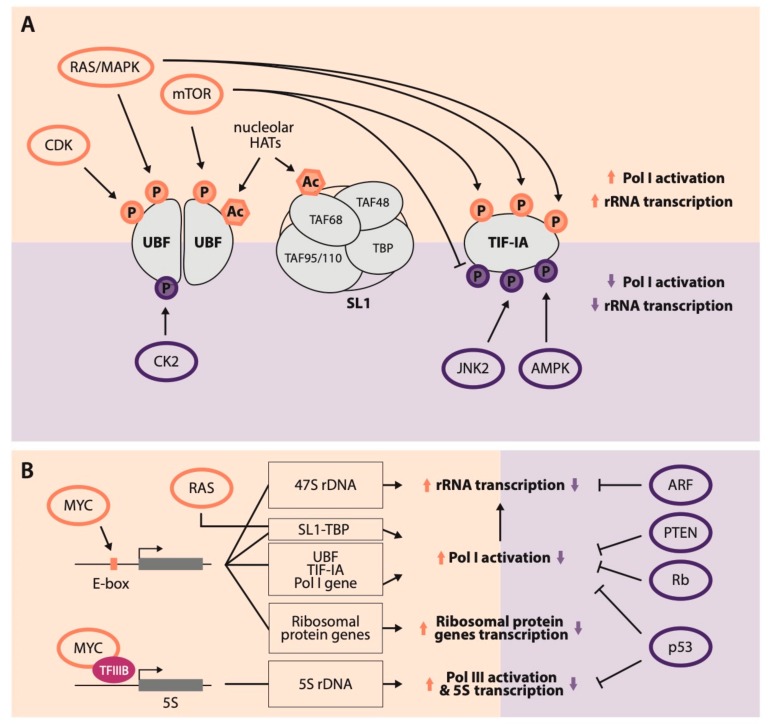
Transcriptional and post-translational regulation of rRNA transcription by oncogenic pathways. (**A**) Schematic representation of the main components of the polymerase I (PolI) pre-initiation complex (PIC) and the major activatory (upper panel, orange) and inhibitory (lower panel, purple) post-translational modifications (PTMs) induced by oncogenes or tumor suppressor genes. (**B**) Schematic representation of the convergent regulatory pathways that boost rRNA transcription upon oncogene activation (left panel, orange) or repress it through tumor suppressor genes (right panel, purple).

**Figure 2 cells-08-01098-f002:**
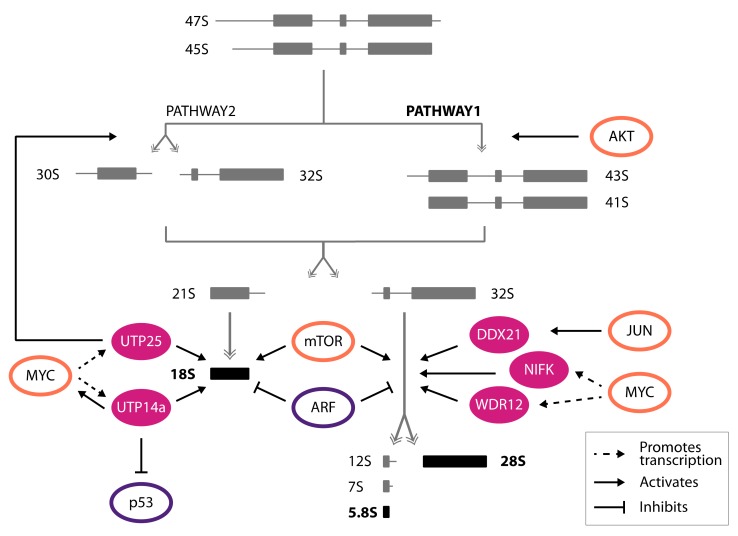
Regulation of rRNA processing in cancer. Schematic representation of the 47S rRNA precursors processing pathways (grey) generating the mature 18S, 5.8S, and 28S molecules (black). The main maturation steps affected by oncogenes (orange ovals) or tumor suppressors (purple ovals) are represented.

**Figure 3 cells-08-01098-f003:**
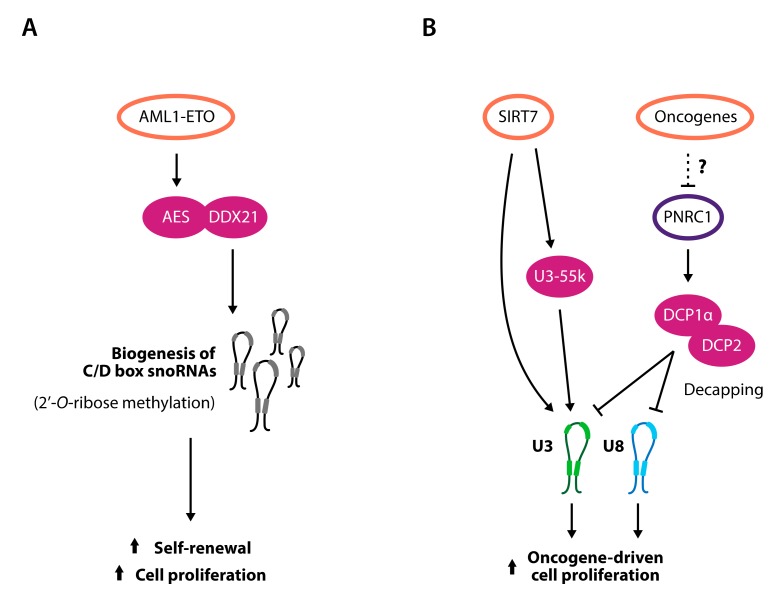
Small nucleolar RNAs (snoRNAs) sustain the oncogenic potential of cancer genes. (**A**) C/D box snoRNAs are required to sustain oncogene-induced self-renewal and proliferation of acute myeloid leukemia (AML) cells. (**B**) U3 and U8 snoRNAs are targets of oncogenes (orange ovals) and tumor suppressors (purple ovals) and are essential to promote oncogene-induced cell proliferation. Oncogene stimulation induces U3 stabilization and putatively prevents U3 and U8 decapping, mediated by a DCP1α/DCP2 complex, tethered inside nucleoli by PNRC1 tumor suppressor.

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
