# Peer review of "How Cancer Exploits Ribosomal RNA Biogenesis: A Journey beyond the Boundaries of rRNA Transcription"

_cells, 2019, doi:10.3390/cells8091098_

Round 1

Reviewer 1 Report

In the present review the Authors, after a brief introduction on the mechanism of ribosome biogenesis and nucleolar organization, present how oncogenic and tumor suppressive pathways regulate rRNA transcription and processing.

The work is very well organized and provides a clear view on the impact of major oncogenes and tumor suppressors on the regulation of ribosome biogenesis.

Below are listed few suggestions the Authors may take into account to improve the manuscript:

The tumor suppressor gene ARF is also known to control rRNA transcription (and not only processing) acting on UBF and TTF-1, see for instance Ayrault et al. Oncogene 2006 and Lessard et al. Molecular Cell 2010. When  alterations of snoRNA are presented the authors focus in particular on their role on rRNA processing. It should be mentioned if they lead to altered rRNA modification and how this may play a role in cancer.

minor points:

page 2 line 61 "ITS1-2 and ETS1-2" it's not clear, I suggest to explicitate ITS1 and ITS2 and ETS1 and ETS2

Reviewer 2 Report

In this manuscript, the authors present a review on links between control of rRNAs production and tumorigenesis. Overall, the review present interesting data that link oncogenic signals, such as c-myc, to ribosomes production in a well focus manner. Then, the authors present their own data on the control of rRNA modifications by snoRNAs in the context of cancer cells. This is very innovative and highlight a new interesting pathway of tumour cell transformation. Although this review is globally very interesting and well written, I would suggest some changes before publication to increase scientific accuracy.

Here, are my more specific comments: 

-Concerning p53 regulation, since only L5,L11 in association with 5S rRNA have been demonstrated to be required for p53 stabilisation in answer to ribosomal stress, I would suggest to precise these data in the text (lanes 150 to 153).

-Figure 2, is not very clear. Both UTP25 and UPT14a act mainly on 90S pre-ribosomes and primarily alter early processing of 47/45S although 30S maturation to 21S is also affected. Therefore, I would suggest to displace those proteins from the transition between 21S to 18S-E. Overall, the clarity of this figure could be improved as the take-home message is not clear.

minor comments:

-L31: « the assembly of ribosome subunits » could be misleading for the reader that could think about ribosome subunit joining. In that context I would suggest to use « the assembly of ribosomal subunits ».

-L46: I think many of the RNA helicases that intervene during ribosome maturation, have been well caraterised for supporting RNA helicase activity both in vivo or in vitro, therefore I would suggest to take out “putative” in this context.

-L250: There are not so many snoRNAs involved in processing instead or in top of rRNA modifications. I think it would be interesting to indicate the number of snoRNAs involved in modifications and named all the ones involved in processing : SNORD3,SNORD118, SNORD17, SNORD14, SNORA73A and SNORA21.
